# Attenuation correction using 3D deep convolutional neural network for brain [18]F-FDG PET/MR: Comparison with Atlas, ZTE and CT based attenuation correction

**Paul Blanc-Durand**[1]*, **Maya Khalife**[2], **Brian Sgard**[1], **Sandeep Kaushik**[3], **Marine Soret**[1], **Amal Tiss**[4], **Georges El Fakhri**[4], **Marie-Odile Habert**[1,5], **Florian Wiesinger**[6], **Aurélie Kas**[1,5]

**1** Nuclear Medicine Department, Groupe Hospitalier Pitié-Salpêtrière C. Foix, APHP, Paris, France, **2** Centre de Neuroimagerie de Recherche (CENIR), Institut du Cerveau et de la Moëlle, Paris, France, **3** GE Healthcare, Bangalore, India, **4** Gordon Center for Medical Imaging, Radiology, Massachusetts General Hospital, Harvard Medical School, Boston, Massachusetts, United States of America, **5** Laboratoire d'Imagerie Biomédicale, Sorbonne Université, Paris, France, **6** GE Healthcare, Munich, Germany

* paul.blancdurand@aphp.fr

## Abstract

One of the main technical challenges of PET/MRI is to achieve an accurate PET attenuation correction (AC) estimation. In current systems, AC is accomplished by generating an MRI-based surrogate computed tomography (CT) from which AC-maps are derived. Nevertheless, all techniques currently implemented in clinical routine suffer from bias. We present here a convolutional neural network (CNN) that generated AC-maps from Zero Echo Time (ZTE) MR images. Seventy patients referred to our institution for [18]FDG-PET/MR exam (SIGNA PET/MR, GE Healthcare) as part of the investigation of suspected dementia, were included. 23 patients were added to the training set of the manufacturer and 47 were used for validation. Brain computed tomography (CT) scan, two-point LAVA-flex MRI (for atlas-based AC) and ZTE-MRI were available in all patients. Three AC methods were evaluated and compared to CT-based AC (CTAC): one based on a single head-atlas, one based on ZTE-segmentation and one CNN with a 3D U-net architecture to generate AC maps from ZTE MR images. Impact on brain metabolism was evaluated combining voxel and regions-of-interest based analyses with CTAC set as reference. The U-net AC method yielded the lowest bias, the lowest inter-individual and inter-regional variability compared to PET images reconstructed with ZTE and Atlas methods. The impact on brain metabolism was negligible with average errors of -0.2% in most cortical regions. These results suggest that the U-net AC is more reliable for correcting photon attenuation in brain FDG-PET/MR than atlas-AC and ZTE-AC methods.

**Data Availability Statement:** All relevant data are within the manuscript and its Supporting Information files.

**Funding:** The funder provided support in the form of salaries for authors [MK, SK, FW], but did not have any additional role in the study design, data collection and analysis, decision to publish. The specific roles of these authors are articulated in the 'author contributions' section.

**Competing interests:** MK received a research grant from GE Healthcare. SK and FW are GE Healthcare employees. Only non-GE employees had control of inclusion of data and information that might present a conflict of interest for authors who are employees of GE Healthcare. No other potential conflict of interest relevant to this article was reported. AK received honoraria for lectures from GE Healthcare and Piramal. MOH received honoraria for lectures from Lilly. This does not alter our adherence to PLOS ONE policies on sharing data and materials. The other authors do not present any conflict of interest.

## Introduction

An important technical challenge of integrated PET/MR is achieving an accurate PET attenuation correction (AC) comparable to transmission-based computed tomography (CT). In current PET/MR scanners, AC is usually accomplished by generating an MRI-based surrogate CT from which AC-maps are derived [1]. Commercially available MR-based AC methods (MRAC) include a 2-point Dixon MR that is segmented into four tissue classes (e.g. lung, air, fat and soft-tissue) in which predefined attenuation coefficients are assigned [2]. In brain PET imaging, it is supplemented by the use of a single-atlas generated from an average of normal CT scans (atlas-based AC), or replaced by an ultrashort or zero echo time (UTE/ZTE) MRI to capture air and bone information. In the former, the single atlas is warped and co-registered to each new subject. In the latter, discrete coefficients are assigned to classes of voxels segmented through image processing or, more recently, through machine learning [3].

Previous studies using atlas-based techniques have reported limitations when applied to unusual head anatomies and highly dependency on image registration quality of CT templates to individual MR images. ZTE-based AC has been previously proposed as a good candidate for accurate and subject-specific AC [4]. Recently, this method was improved with shorter acquisition time making it compatible with clinical settings, and demonstrated promising performances compared to the atlas-AC in a small number of subjects [5].

Deep learning (DL) has witnessed increased interest in medical imaging and was recently employed to generate pseudo CT from brain MR images to correct PET for photon attenuation. To date, DL-based AC methods have been evaluated in relatively small cohorts, mainly focusing on the quantitative accuracy of the predicted CT. Liu et al. trained a 2D convolutional auto encoder from T1 weighted MR images to segment MRI in three-tissue classes (bone, soft tissue, air) and evaluated the errors on brain PET image reconstruction in 10 patients [3]. Gong et al [6] reported the superiority of a 2D convolutional neural network (CNN) to generate brain AC maps either using Dixon MRI alone or combined with ZTE MRI as input data, over Dixon- and segmented-ZTE based techniques in a sample of 14 patients. In the same vein, Leynes et al. tested a deep CNN with combined ZTE and Dixon MRI for pelvic PET/MRI attenuation correction and showed that quantitative assessment of lesions was improved compared with the standard Dixon-based method [7]. More recently, Ladegofed et al [8] found that their UTE based CNN was visually more appropriate and improved quantification accuracy compared to UTE in a cohort of pediatric brain tumors regardless of age.

In this work, we propose to test a 3D U-net CNN to compute AC maps from ZTE MR images for brain FDG-PET/MR attenuation correction in a large cohort of patients referred for cognitive impairment investigation. Our aim is to compare this approach with the reference CTAC, and with clinically available MRAC techniques.

## Materials and methods

### Population

Seventy consecutive patients (68.2 ± 13.7 y/o, 37 men) examined at our institution between July 2016 and December 2017, and fulfilling the following criteria, were included: 1) brain FDG-PET/MR scan (SIGNA PET/MR, GE Healthcare, Waukesha, WI, USA) performed in the context of neurocognitive disorder investigation; 2) available brain CT scan without iodinated contrast agent injection; 3) no skull, face, neck surgery or injury between CT and PET/MR examinations. 23 of the 70 patients were added to the training set of the manufacturer which comprised a total of 50 patients from 4 institutions (all equipped with the Signa PET/MR). The 47 others were only used in the validation set. All studies were performed on the PET/MR

scanner at the Pitié Salpètèriêre Hospital, Paris, France, within routine hospital appointments and data use was approved by the French authority for the protection of privacy and personal data in clinical research (Commission Nationale de l'Informatique et des Libertés, approval No. 2111722). This study was performed according to the principles of the Declaration of Helsinki.

## PET/MR acquisition

A dose of 2 MBq/kg of FDG was injected 30 to 45 min prior to PET/MR acquisition. Patients rested in quiet surroundings with their eyes closed for at least 20 min post-injection. PET acquisition of the head lasted 20 minutes and was performed while MR images were acquired using an 8-channel brain coil, including 3D T1-weighted inversion-recovery fast spoiled gradient echo acquisition, 3D (Fluid Attenuation Inversion Recovery) FLAIR, 3D susceptibility-weighted MRI and axial diffusion-weighted MRI. A two-point Dixon (Liver Acquisition with Volume Acceleration: LAVA-Flex in the manufacturer's nomenclature) T1-weighted MRI was acquired yielding in-phase and out-phase images, from which water and fat weighted images were calculated. The following parameters were used: axial acquisition; TR = 4 ms; TE = 1.12 and 2.23 ms; flip angle 5˚; slice thickness 5.2 mm with a 2.6 mm overlap; 120 slices; pixel size of 1.95x1.95 mm$^2$. Additionally, a proton density weighted ZTE MRI was acquired with the following parameters: 3D center-out radial acquisition; voxel size 2.4x2.4x2.4 mm$^3$; field of view (FOV) 26.4x26.4 cm$^2$; flip angle 0.8˚; bandwidth ± 62.5 kHz; TR = 390 ms; TE = 0 ms; acquisition time 40 s.

## ACmaps generation

Four AC maps were generated for each patient with CTAC map set as reference method. First, an estimation of the Hounsfield's Units (HU) was performed for each AC method followed by a post-processing pipeline to include external material (table, coils) and convert HU to linear attenuation coefficients in cm$^{-1}$. For CTAC, helical CT scans were collected from previous brain PET/CT examinations. CTAC maps were created by a a non-rigid registration of the patient CT to the ZTE MRI using a mutual information maximization algorithm with Statistical Parametric Mapping (SPM12) software (http://www.fil.ion.ucl.ac.uk/) with 4[th] degrees B-spline interpolation. The time interval between CT and ZTE co-registred PET/MR scans was 13.3 ± 18.7 months. Between this time-lapse, no surgery or other interventions affecting brain's attenuation were observed.

Next, an atlas-AC map (AC map$_{Atlas}$) was created by an elastic registration of an average CT template on the Dixon in-phase image using the implemented process on the PET/MR scanner [9]. A third AC map was generated from ZTE MRI segmentation (AC map$_{ZTE}$). An offline post-processing pipeline provided by the manufacturer, was used to generate bone tissue estimates from the acquired ZTE data based on the method proposed by Wiesinger et al [10] and whose version included sinus-edge correction as described in [11, 12]. In this approach, voxels classified as bone were assigned density values of 42 + 2400 (1 − IZTE) HU, where IZTE represents the normalized ZTE intensity. Additionally, using morphological operations, the paranasal sinuses were detected and masked to avoid misclassification of air to bone. Finally, an AC map was created by the deep CNN, as described below, and is referred to as AC map$_{U-net}$.

**CNN design and training.** From the ZTE volumes in the training set, normalized from mean and standard deviation, patches of 64x64x16 were randomly picked and used for training. A 3D U-net architecture was used [13, 14] and can be seen in S1 Fig. The downward path consisted of four stages of batch normalized 3D convolutions activated by the exponential

linear unit function and connected by maxpooling operations with respectively 8,32,32,8 filters of kernel size $3^3$. The upward path mirrored the first four layers and was connected to the latter by concatenation. Model weights (130000 parameters) evolved by reducing the mean squared error between network output and reference CT image via the Adam optimizer. No data augmentation was performed, the 3D patched provided implicit augmentation for structure invariance. Model was trained for 50 000 iterations of batch size 100. Algorithm was implemented in Keras on a NVIDIA V100 GPU. Inference time for a whole volume with 75% overlap between patches was for one patient about 10s.

**Post-processing of AC maps.** All Atlas-, ZTE-, Unet-, CT-AC maps were then processed as follows using an in-house program developed in MATLAB (MathWorks Inc., Natick, MA, USA). Coils, bed and all surrounding material were added to AC maps. They were then converted from HU to linear attenuation coefficients using a bi-linear transformation taking into account the difference of attenuation from annihilation's photons at 511 keV and CT tube energy at 120kV [15]. All AC maps were smoothed with a Gaussian filter with full-width at half-maximum of 10 mm, which is the default smoothing applied on the atlas-AC map by the manufacturer.

## PET reconstruction

PET images were reconstructed with every AC-map using projected sinogram PET data with the following reconstruction parameters: Ordered Subsets Expectation Maximization (OSEM) algorithm with time of flight (TOF), 8 iterations and 28 subsets and with a transaxial post-reconstruction Gaussian filter of 3 mm, resolution correction with point spread function (PSF) modeling, attenuation and scatter corrections, FOV of 300 x 300 mm$^2$, voxel size of 1.17x1.17x2.78 mm$^3$, with 89 slices along z. Finally, a set of 4 corrected PET images (PET$_{Atlas}$, PET$_{ZTE}$, PET$_{U-net}$, and PET$_{CT}$) was obtained for each patient.

## Data analysis

The Dice Similarity Coefficient (DSC) was used to assess the accuracy of bone information provided by ZTE- and U-net AC maps. For this, bone masks were computed from MRAC maps and CTAC map, by excluding voxels with intensity lower to 200 HU. DSC was computed as follows:

$$\text{Dice Similarity Coefficient} = \frac{2 \times \text{MRAC} \cap CTAC}{\text{MRAC} + \text{CTAC}}, \tag{1}$$

where MRAC corresponds to the bone mask obtained by thresholding the different MR-based AC maps and, CTAC is the bone mask obtained by thresholding reference CTAC maps. Statistical difference of DSC between AC methods was assessed using paired t-test. Next, to assess the impact of AC method on brain PET brain and standard uptake value (SUV) measurements, the relative PET error ($\Delta$SUV in %) was calculated as:

$$\text{Relative PET error}: \Delta\text{SUV} = \frac{\text{PET}_{\text{MRAC}} - \text{PET}_{\text{CT}}}{\text{PET}_{\text{CT}}} * 100, \tag{2}$$

where PET$_{\text{MRAC}}$ is the PET image reconstructed using the different AC maps and PET$_{\text{CTAC}}$ is the reference PET image reconstructed using the ground-truth CT. Relative errors in SUV and joint histograms between the different PET$_{\text{MRAC}}$ and PET$_{\text{CTAC}}$ were analyzed after brain masking in the patient native space. Similarity between PET images was estimated using the coefficient of determination R2 and Root Mean Square Error (RMSE). Moreover, median and

interquartile range (IQR = IQ75-IQ25) of ΔSUV were calculated in all voxels and distribution of the latter was depicted through a histogram.

For voxel-based analyses, PET images were spatially normalized in the MNI space with SPM12 using non-linear spatial transformations between their co-registered T1 weighted MR volume and the T1 MNI template. They were then segmented by applying a grey matter mask from the tissue probability map (TPM) template available in SPM. ΔSUV maps (%) were computed at the voxel-level from spatially normalized PET images on the 47 patients for each AC method, with the formula described above. Next, relative PET errors were calculated in 70 volumes of interest (VOIs) extracted from the automated anatomical labeling atlas (AAL, http://cyceron.fr/freeware). Mean and standard deviation of ΔSUV across patients for each AC method were computed in those regions. Statistical significance was determined using a repeated measures analysis of variance (ANOVA) with a post-hoc paired t-test after the homo-elasticity hypothesis was confirmed using Levine test. All data were processed using python 3.5 with numpy, pandas, matplotlib for data visualization and nilearn for brain masking and plotting [16].

## Results

Example of axial slices of ZTE, CT images and AC maps are shown in Fig 1. Visual assessment reveals no heterotopic calcifications on AC maps generated with U-net method, unlike maps computed from ZTE segmentation. The mean DSC in the validation set between bone structures was $0.786 \pm 0.05$ on ACmap$_{ZTE}$ compared with CT, and $0.81 \pm 0.03$ between ACmap$_{U-net}$ and CTAC maps ($p < 0.05$). Thoses results and Jaccard coefficients are summarized in S1 Table.

Joint histograms between PET images corrected with CTAC and MRAC methods are displayed in Fig 2. All MRAC techniques yielded a voxel distribution around the identity line. Goodness of fit, as measured with both the coefficient of determination R2 and RMSE that reflects how well uptakes were estimated by the different AC methods, was better with the U-net compared to both ZTE- and Atlas-AC. $R^2$ and RMSE were respectively 0.98 and 534.7 for PET$_{Atlas}$ vs. PET$_{CTAC}$, 0.99 and 340.5 for PET$_{ZTE}$ vs. PET$_{CTAC}$, 1.0 and 253.5 for PET$_{U-net}$ vs. PET$_{CTAC}$.

U-net AC provided the most accurate PET quantification, the lowest bias on SUV measurements and the lowest inter-individual variability, compared to ZTE-AC. In native space, median and interquartile ranges (IQ75%-IQ25%) for the bias were $-1.3 \pm 13.5\%$ for atlas-AC, $-3.0 \pm 6.9\%$ for ZTE-AC and $-0.2 \pm 5.6\%$ for U-net-AC. The distribution of PET errors in the native space is shown in Fig 3. Mean ΔSUV maps are shown in Fig 4.

The ANOVA results are displayed in Table 1 and revealed that AC methods, metabolism among cortical regions and their interactions were statistically associated. Post-hoc paired t-test and results of VOI analysis are displayed in S2 Table and S2 Fig. When applying the atlas based-AC, the cortical metabolism was significantly overestimated in regions located above the anterior commissure-posterior commissure (AC-PC) line, reaching a maximum bias in the superior parietal cortex ($4.7 \pm 7.1\%$, $p < 0.05$), the superior occipital cortex ($4.9 \pm 6.6\%$) and the precuneus ($3.7 \pm 4.7\%$), whereas a metabolism underestimation was observed in areas below this line, mainly in the temporal poles ($-6.6 \pm 5.7\%$), the orbito-frontal cortex ($-5.5 \pm 5.0\%$), the inferior temporal cortex ($-7.5 \pm 5.0\%$) and the cerebellum ($-5.2 \pm 7.3\%$, all $p < 0.05$). With ZTE-AC method, a global metabolism underestimation was found, most pronounced in the prefrontal dorsolateral cortex around -3%, except in some subregions of the cerebellum that were over-estimated, reaching a mean error of $0.07 \pm 6.1\%$ in this structure. This difference was not statistically significant. In contrast, U-net AC slightly overestimated the metabolism in regions close to the vertex and in the cerebellum. The bias ranged from

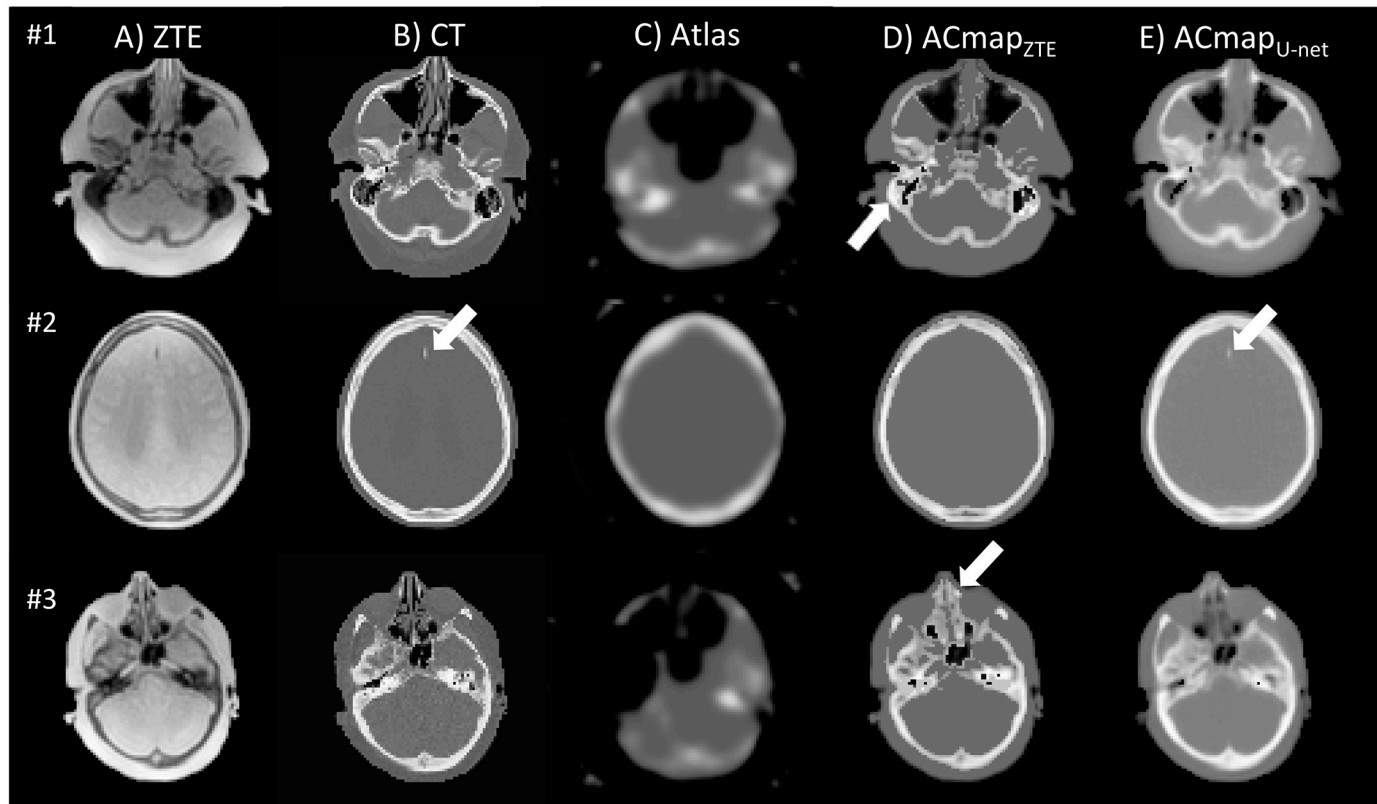

**Fig 1. Axial sections of ZTE MR image (A), CT that served as reference (B), AC map generated from Atlas (C), AC map generated from segmented-ZTE (D), U-net method (E) in three different patients.** With ACmap ZTE (patient #1, ZTE-based AC yields to a global overestimation of attenuation in the ethmoids or mastoids cells as shown in D (arrow). In patient #3, ZTE-based AC yields to a global overestimation of attenuation in the ethmoids or mastoids cells as shown in D (arrows). With ACmapU-net, (E) mastoids cells were often filled with blurry structures. The calcification of a small meningioma of the falx cerebri found in patient #2 is partly restored with the ACmapU-net (arrow) opposite to ZTE- AC.

1.7 ± 2.6% in the right parietal superior cortex to -1.8 ± 1.9% in the left amygdala (all $p < 0.05$). The lowest interregional variability reflected by the standard deviation of mean errors among all regions, was found with the U-net AC method reaching 0.79% compared to 3.13% for Atlas-AC and 0.86% for ZTE-AC.

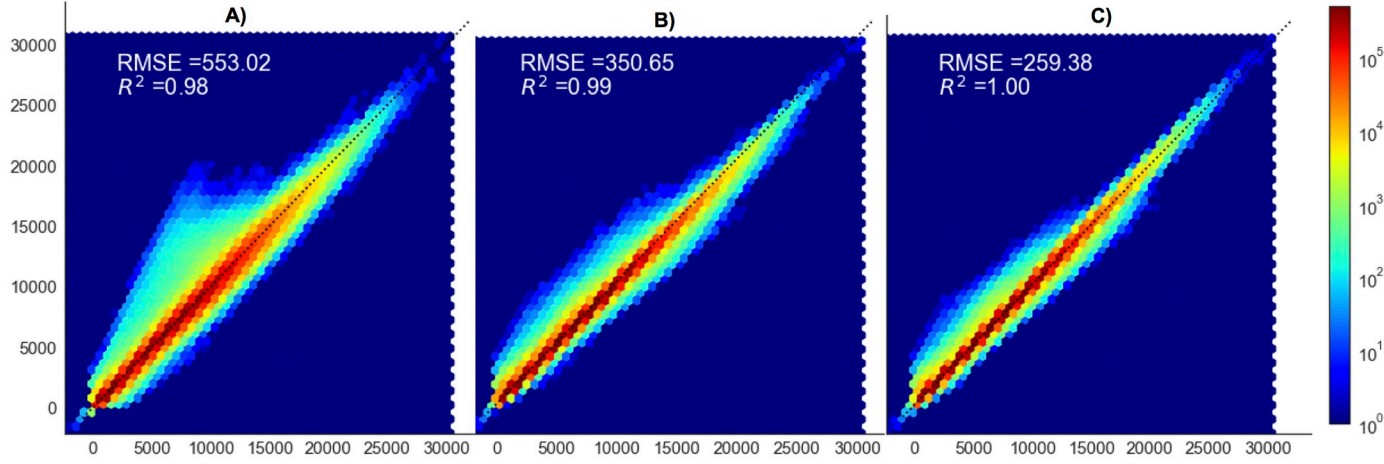

**Fig 2. Joint histograms of PET$_{Atlas}$, (A), PET$_{ZTE}$ (B), PET$_{U-net}$ (C), compared to PET$_{CT}$ before spatial normalization, after brain masking.**

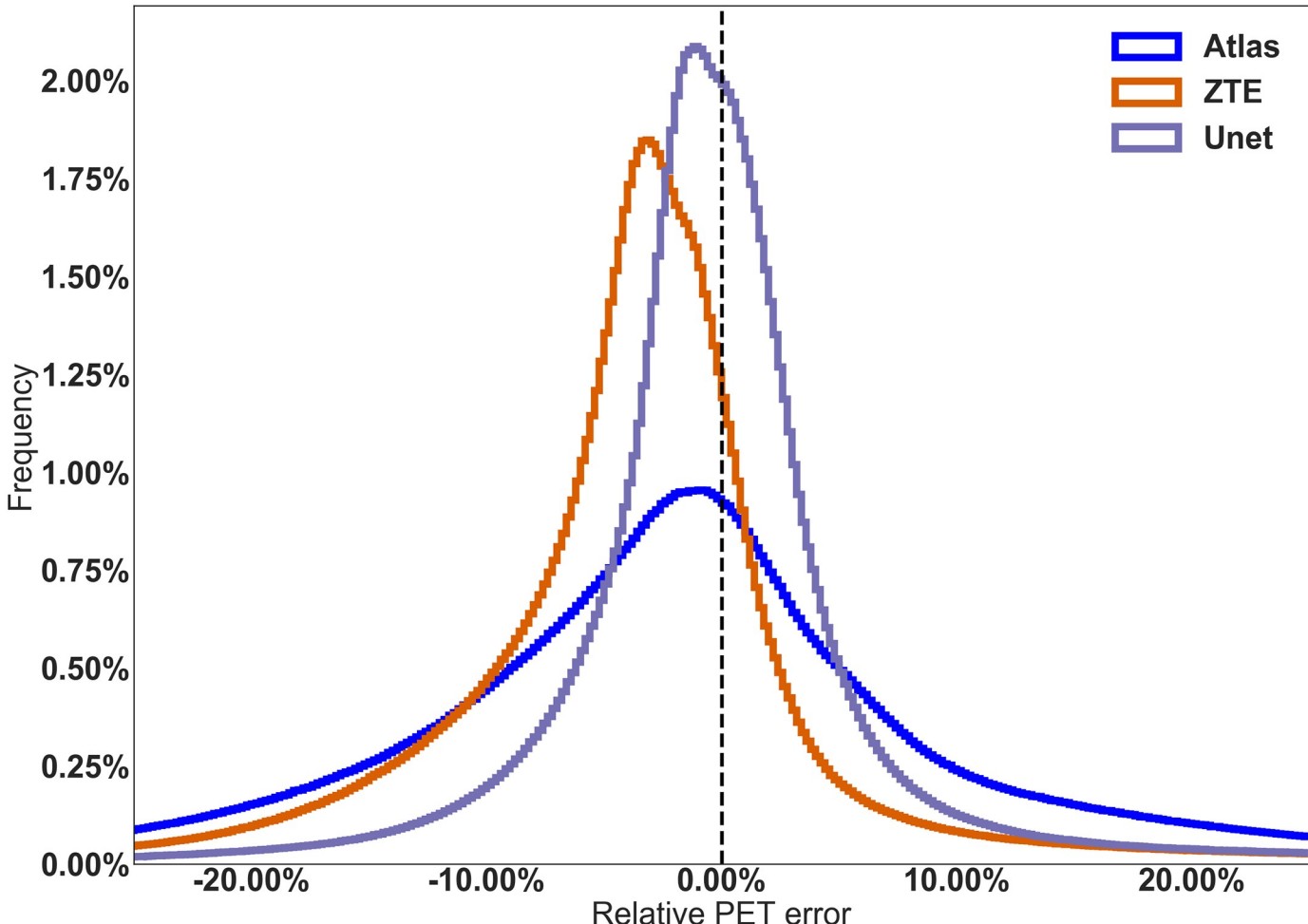

**Fig 3. Distribution of the relative PET errors generated by the U-net AC (dashed lines), the ZTE-based AC (circles) and the atlas-based AC (stars) before spatial normalization, after brain masking.** Along x-axis: relative PET error (ΔSUV) and along y-axis: frequency.

## Discussion

In this study, we evaluated the performances of a deep CNN algorithm with a 3D U-net architecture based on ZTE sequences for AC map estimation in 47 patients explored for cognitive impairment by FDG PET/MRI and compared them to CT scan. Our results indicate that U-net-AC yields the lowest error, as well as the lowest inter- and intra-patient variability compared to the Atlas and the ZTE-based AC methods.

We observed a relative PET error distribution with Atlas and ZTE AC similar to the one previously published in the literature [4, 9, 17]. The Atlas AC overestimates brain metabolism above the AC-PC line especially in the regions close to the vertex whereas it underestimates it in the lower regions including the cerebellum. When using the ZTE-based AC, a mild underestimation (around 2.5%) was found in the overall cortex excluding the cerebellum. This metabolism underestimation can be explained by segmentation errors mainly in regions with air-bone interfaces (for examples, in mastoids, ethmoid bones or frontal sinus). Another valid explanation is partial volume effect of the ZTE MR volumes that can lead to incorrect assignments of air or soft tissue into bone [4, 5]. It is nevertheless interesting to see that those errors were not observed when absolute quantification was performed [18].

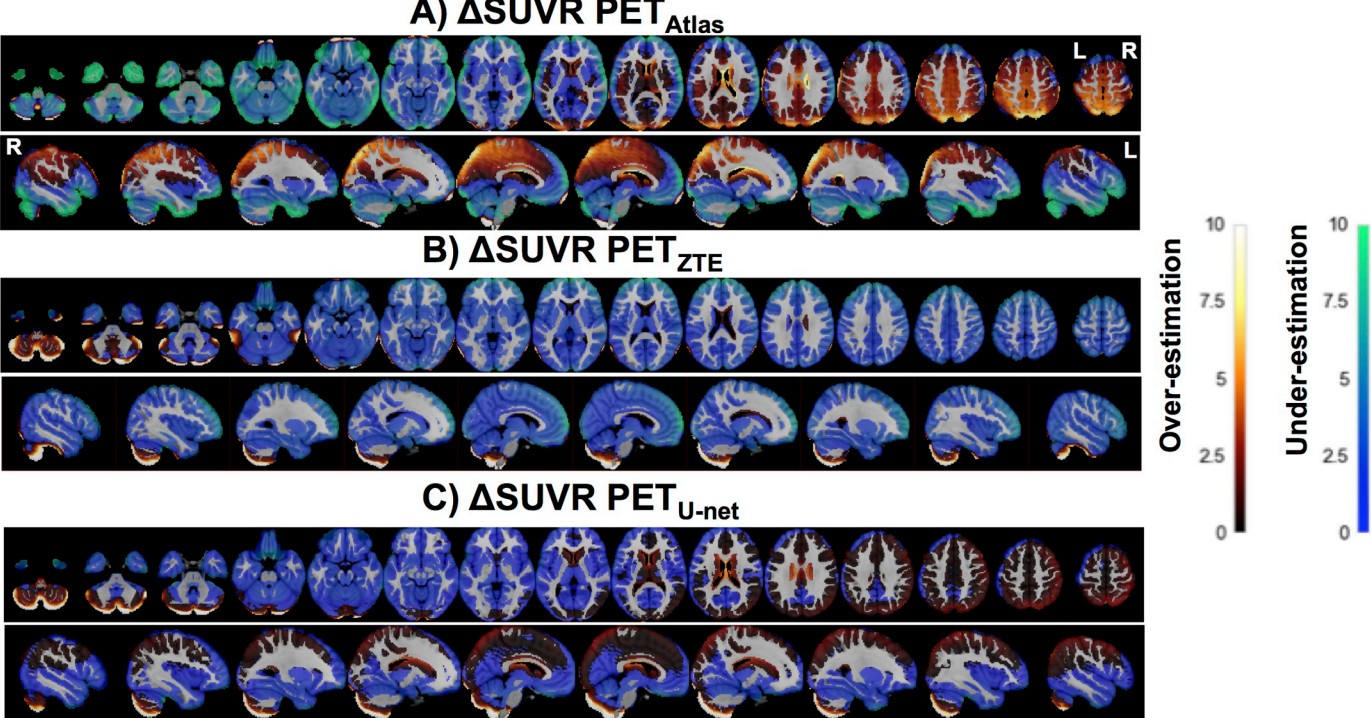

**Fig 4. Axial slices of the PET error map for the different AC methods Atlas (A), ZTE (B) and U- net (C) compared to CTAC.**

The brain distribution of U-net AC errors reveals a very small error (with mean -0.2% and ranging from maximum 1.7% close to the vertex to -1.8% in the temporal lobe). VOI analyses and difference maps generated from the corrected PET images confirmed the superior performance of the AC method using a 3D U-net architecture over the other methods as it provided a very low bias and lower inter individual and inter regional variability compared to ZTE- and Atlas-based AC techniques. The cerebellum remains the region with the highest variability in metabolism between individuals probably because of difficulties in correctly estimating density of mastoid cells.

Our deep CNN revealed similar performance as Gong et al [6] who were the first to evaluate two deep CNN architectures using both ZTE and Dixon MRI for brain AC estimation. In fact, for the bone segmentation procedure at the same threshold of 200 HU, our U-net reached a DSC of 0.80, that was statistically significantly improved compared to ZTE and that was in the same order of magnitude compared to 0.77 for their U-net and 0.80 for their Grouped U-net. They also found that the U-net architecture with ZTE as input, performed better than segmentation based ZTE AC or other MRAC approaches, yielding less bias and variability with a mean whole brain metabolism over-estimation around 2%. However, our study revealed slight difference in the error distribution that can be explained by multiple

**Table 1. Results of the analysis of variance (ANOVA) before the post-hoc paired t-test analysis.**

| Factors | F-score | p-value | eps |
|---|---|---|---|
| Attenuation Correction Methods | 12.72 | 4.56e-05 | 0.56 |
| AAL regions | 47.87 | 1.73e-52 | 0.11 |
| Interaction | 43.56 | 2.38e-38 | 0.02 |

methodological points. Indeed, our larger population allowed to use a correct validation cohort and to benefit from the contextual information provided by a 3D deep CNN that could not be performed in their case. Also we chose to use ZTE only as input whereas they used both DIXON and ZTE.

Neurodegenerative diseases lead to synaptic losses and neuronal dysfunctions. In Alzheimer disease, cortical hypometabolism [19] usually starts in the precuneus and posterior cingulum areas and later extends to parieto-temporal associative areas, and more occasionally in the frontal or occipital associative areas [20]. The Atlas AC particularly impacted those areas where an over-estimation of the regions in parietal, occipital and temporal associative cortex could result in a loss of sensitivity. Conversely, with the U-net AC method, metabolism in the cingulum posterior or precuneus areas was not statistically different from CTAC. Therefore brain AC with the U-net method is preferable in the context of neurodegenerative diseases.

Regarding the input that is fed to the CNN, most studies include MR volumes or slices, essentially from morphological MRI such as T1 or T2 weighted MR sequences that are routinely performed for the exploration of neurodegenerative disorders or glial tumors. As the signal in the bone structures is low, mapping complex functions to predict HU from those inputs may requires more data for efficient training. This is one of the reasons why some authors used specialized pulse sequences such as UTE or ZTE that provide additional air and bone information [6, 7, 21] Nevertheless, we can't be sure that the CNN is actually using the bone information contained in ZTE to make its prediction. Main alternatives to MR inputs are the PET data themselves that can also be fed to the CNN. For example, using the maximum likelihood reconstruction of activity and attenuation (MLAA) augmented by TOF which allows at the same time simultaneous reconstruction of activity and attenuation using PET emission data, Hwang et al used the AC map produced by MLAA as an entry of the CNN to predict CT with good accuracy reaching a DSC of 0.79 [22].

Our study, even if was separated into a training and testing sets and included a relatively large sample of patients compared to already published studies of patients, remains a pretty small cohort in the era of big data. One potential limitation was the use of brain CT scans acquired on different systems. However, spatial normalization in the MNI space and Gaussian smoothing of 10 mm applied to all AC-maps reduced this variability. That Gaussian filter is nevertheless questionable as it may cancels out subtle differences that the CNN method was actually trying to model. Another potential limitation was the delay between PET/MRI and CT. However, a visual examination between CT and MRI was systematically performed to ensure that no morphological alteration had appeared during this time interval. It is to note that as on no major morphological skull or brain abnormalities were present in the training nor in the validation datasets, it is difficult to infer how the CNN will behave in those conditions. Nevertheless, the small falci meningioma visible in Fig 1 is reassuring. Also, a previous study [8] including abnormal brain MRI with history of surgery and presence of titanium coils in 33% of patients which resulted to signal voids were well recovered by their deep learning algorithm. Lastly, similarly to most of the existing deep neural networks for AC estimation, we tried to predict CT from MRI. Indeed CT was used both for training and as the reference method [23]. In a recent multi-center study of several PET/MRI attenuation protocols stated that not comparing to gold standard transmission scans was a limitation [24]. To overcome this limitation, some authors trained a deep neural network to predict [68]Germanium transmission scan instead of CT [25] which lead to similar accuracy compared to the latter. Furthermore, such studies have limited clinical use given that no standalone PET scanners are available today from any of the PET manufacturers.

## Conclusion

We have presented in this work a 3D-U-net architecture to generate the attenuation map to correct brain PET imaging based on the ZTE MR images. The results show that the U-net AC method is more suited for the exploration of cognitive disorders than both atlas-AC and ZTE AC methods as it has the lowest bias as well as the lowest inter and intra patients variability compared to reference CTAC.

## Supporting information

**S1 Fig. Architecture of the deep CNN used for training.** It takes as input a 64x64x16 patch from ZTE that goes through an encoding followed by a decoding path which concatenated by concatenations, and produces a 64x64x16 pseudoCT.
(TIFF)

**S2 Fig. Radar plot of mean (a) and standard deviation (b) of relative PET errors.** Radar plot among patients (n = 47) for Atlas, ZTE and U-net based attenuation correction within the 70 VOIs from AAL template.
(TIFF)

**S1 Table. Dice similarity *(DSC)* and Jaccard coefficients of ZTE and U-net AC maps.**
(TEX)

**S2 Table. VOI mean and standard deviations in the 70 grouped automatic anatomic labeling regions and whole cortical brain for PET$_{Atlas}$, PET$_{ZTE}$ and PET$_{U-net}$.** Significance level of paired t-test are rapported and corrected for multiple comparison at the Bonferroni level as follow: $^{**}$ corrected p-value < 0.01 and $^*$ for corrected p-value < 0.05.
(TEX)

## Author Contributions

**Conceptualization:** Paul Blanc-Durand, Maya Khalife, Aurélie Kas.

**Data curation:** Paul Blanc-Durand, Maya Khalife, Brian Sgard, Aurélie Kas.

**Formal analysis:** Sandeep Kaushik.

**Investigation:** Aurélie Kas.

**Methodology:** Paul Blanc-Durand, Maya Khalife, Marie-Odile Habert, Aurélie Kas.

**Project administration:** Paul Blanc-Durand, Aurélie Kas.

**Resources:** Paul Blanc-Durand, Marine Soret.

**Software:** Sandeep Kaushik.

**Supervision:** Maya Khalife, Aurélie Kas.

**Validation:** Paul Blanc-Durand, Aurélie Kas.

**Visualization:** Paul Blanc-Durand.

**Writing – original draft:** Paul Blanc-Durand, Aurélie Kas.

**Writing – review & editing:** Paul Blanc-Durand, Maya Khalife, Amal Tiss, Georges El Fakhri, Marie-Odile Habert, Florian Wiesinger, Aurélie Kas.

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
