## [Decision Letter · Decision Letter 0]

4 Aug 2019

PONE-D-19-16876

Attenuation Correction using 3D Deep Convolutional Neural Network for Brain 18F-FDG PET/MR: Comparison with Atlas, ZTE and CT based Attenuation Correction.

PLOS ONE

Dear Dr Blanc-Durand,

Thank you for submitting your manuscript to PLOS ONE. After careful consideration by two Reviewers and an Academic Editor, please make the suggested corrections posed by the Reviewers so I can render a decision on this manuscript.

**Comments to the Author**

1. Is the manuscript technically sound, and do the data support the conclusions?

Reviewer #1: Yes

Reviewer #2: Yes

2. Has the statistical analysis been performed appropriately and rigorously? 

Reviewer #1: No

Reviewer #2: Yes

3. Have the authors made all data underlying the findings in their manuscript fully available?

Reviewer #1: Yes

Reviewer #2: No

4. Is the manuscript presented in an intelligible fashion and written in standard English?

Reviewer #1: Yes

Reviewer #2: Yes

5. Review Comments to the Author

Reviewer #1: This is a paper comparing several attenuation correction (AC) methods of PET/MRI images quantitatively. They scanned 70 patients, and 23 of them were used for training of the convolutional neural network (CNN) method for MRAC. They compared individual CT attenuation correction (CTAC) obtained from PET/CT scanners with the MRAC methods using CT-atlas, zero-echo time (ZTE) MR image and CNN with 3D U-net architecture. The regional differences of brain FDG images were less than 5-6 %, and the U-net CNN MRAC method showed the most similar values to the standard CTAC. Although the scan intervals between PET/CT and PET/MR were relatively long, the results are interesting and consistent with several previous reports. There are only a few comments and concerns from this reviewer.

Comments:

1. Page 3/13, PET reconstruction: filter size of OSEM parameter should be added.

2. For multiple comparison performed in the Supplementary Table 2, repeated measures ANOVA with a post-hoc test should be used because the significant difference with Bonferroni correction for 70 regions should be p < 0.05/70, which is almost impossible to show significance.

3. Page 4/12 Results: Fig. 2 shows similar results between ZTE- and U-net MRAC. It is not clear why the authors described that the correlation of U-net is better than ZTE.

4. Figure 3: the distribution curves are overlapped and the individual curve is not clear. Curves without painting colors would be better to observe differences.

5. Recent similar investigations should be compared and discussed such as;

Eur J Nucl Med Mol Imaging 2016; 43: 2190-2200 and EJNMMI Research 2019; 9(1): 16.

Reviewer #2: The paper by Blanc-Durand et al. describes the synthesis and analysis of an improvement on an existing method for attenuation correction (AC) the Zero Echo time. The authors advocate the use of an existing convolution neuro network architecture (3D U-net) to generate improved ZTE-based attenuation maps. I was shown that this method produces better results compared to the actual clinical implementation for PET AC in terms of similarity and relative SUV differences to CTAC, chosen by the authors as the standard for AC.

The usage of a deep learning approach normally involves long waiting times during training, which was circumvent by using a dedicated deep learning data centre GPU. However, for a full deployment as a scanner AC method, the training set would need to see its number at least doubled, also include data for patients with skull abnormalities, e.g. missing bone, 3D printed bone implants, etc…

The proposed method is of high interest to all GE PET/MR users doing brain PET, as well as it opens a precedent for improvement in AC methods for all other PET/MR manufacturers.

Minor comments

1. Line 33: Where is red Ladegofeld et al. should be Ladefoged et al.

2. Line 77 – 80: authors explain registration of CT to ZTE but fail to state which interpolation method (trilinear, b-spline) was used. E.g. CT (512x512x71)-> ZTE (128x128x89).

3. Line 125: Jaccard Index is more sensitive to mismatch. Why did the authors decided to use Dice instead?

4. Having a table for Dice coefficients for the 3 methods, even as supplementary would be relevant for some readers.

5. Adding whole-brain grey matter to S1 Table would provide a better overall understanding of the results.

6. Lines 68, 69,70,121: mm2 and mm3 should be superscript

7. Line 176 and 188: % character is missing in (3.7 ± 4.7) and (0.79, 3.13 and 0.86), respectively.

8. How big is the impact of the calcification detection in a LOR passing though that area, how much is it relevant?

9. Could the authors comment on the performance of the 3D-CNN if a subject shows skull abnormalities (e.g. missing skull)?

6. PLOS authors have the option to publish the peer review history of their article (what does this mean?). If published, this will include your full peer review and any attached files.

**Do you want your identity to be public for this peer review?** For information about this choice, including consent withdrawal, please see our Privacy Policy.

Reviewer #1: No

Reviewer #2: No

We would appreciate receiving your revised manuscript by November, 2019. To enhance the reproducibility of your results, we recommend that if applicable you deposit your laboratory protocols in protocols.io, where a protocol can be assigned its own identifier (DOI) such that it can be cited independently in the future. For instructions see: http://journals.plos.org/plosone/s/submission-guidelines#loc-laboratory-protocols

We look forward to receiving your revised manuscript.

Kind regards,

Stephen D. Ginsberg, Ph.D.

Section Editor

PLOS ONE

1. Please include a copy of Table 1 which you refer to in your text on page 5

2. Please ensure that you refer to Figures 1-4 in your text as, if accepted, production will need this reference to link the reader to the figure.

---

## [Author Response · Author response to Decision Letter 0]

10 Sep 2019

PONE-D19-16876

Enclosed is the revised version of our manuscript, entitled “Attenuation Correction using 3D Deep Convolutional Neural Network for Brain 18F-FDG PET/MR: Comparison with Atlas, ZTE and CT based Attenuation Correction.” that we would like you to reconsider for publication in PlosOne journal. 

We would like to thank the Editorial Committee for reviewing our manuscript and for their relevant and constructive comments. We believe that addressing the raised questions have substantially improved the quality of our manuscript.

In the following, we provide a detailed answer to each of the Reviewers’ comment, and indicate in RED the corresponding changes made in the revised manuscript.

Sincerely Yours,

Paul Blanc-Durand, MD

Corresponding author

  Response to reviewer 1 

1. Page 3/13, PET reconstruction: filter size of OSEM parameter should be added.

We have added the filter size of OSEM. 

Manuscript was updated as followed P3/13: “PET images were reconstructed with every AC-map using projected sinogram PET data with the following reconstruction parameters: Ordered Subsets Expectation Maximization (OSEM) algorithm with time of flight (TOF), 8 iterations and 28 subsets and with a transaxial post-reconstruction Gaussian filter of 3 mm, resolution correction with point spread function (PSF) modeling, …”

2. For multiple comparison performed in the Supplementary Table 2, repeated measures ANOVA with a post-hoc test should be used because the significant difference with Bonferroni correction for 70 regions should be p < 0.05/70, which is almost impossible to show significance.

We thank the reviewer for this remark. We added the ANOVA test in both the methodology part and in the results. Homoelasticity was checked using the Levin test and the latter authorized us to use an ANOVA to check mean differences between groups. 

P4/L154: “Statistical significance was determined using a repeated measures analysis of variance (ANOVA) with a post-hoc paired t-test after the homoelasticity hypothesis was confirmed using Levin test.”

And P5/L176 : The ANOVA results are displayed in table 1 and revealed that AC methods, metabolism among cortical regions and their interactions were statistically associated.

 3. Page 4/12 Results: Fig. 2 shows similar results between ZTE- and U-net MRAC. It is not clear why the authors described that the correlation of U-net is better than ZTE.

Thanks for pointing this. The superiority of the U-net methods for bone prediction and correlation on a voxel to voxel basis was made more on the quantitative part both with RMSE and R2 coefficient. 

To make it clearer manuscript was updated as follow P4/L163 : Goodness of fit, as measured with both the coefficient of determination R2 and RMSE that reflects how well uptakes were estimated by the different AC methods, was better with the U-net compared to both ZTE- and Atlas-AC. R2 and RMSE were respectively …

 4. Figure 3: the distribution curves are overlapped and the individual curve is not clear. Curves without painting colors would be better to observe differences.

We thank the authors for this relevant remark. Hatching and painting were removed in figure 3 for more clarity. Now figure 3 looks like the following.

 5. Recent similar investigations should be compared and discussed such as; Eur J Nucl Med Mol Imaging 2016; 43: 2190-2200 and EJNMMI Research 2019; 9(1): 16.

Those two articles were discussed in the discussion.

 P7/L214 (for Eur J Nucl Med Mol Imaging 2016; 43: 2190-2200 ) and This is one of the reasons why some authors used specialized pulse sequences such as UTE or ZTE that provide additional air and bone information.

P7/L208 for (and EJNMMI Research 2019; 9(1): 16): It is nevertheless interesting to see that those errors were not observed when absolute quantification was performed.

Response to Reviewer #2

The paper by Blanc-Durand et al. describes the synthesis and analysis of an improvement on an existing method for attenuation correction (AC) the Zero Echo time. The authors advocate the use of an existing convolution neuro network architecture (3D U-net) to generate improved ZTE-based attenuation maps. I was shown that this method produces better results compared to the actual clinical implementation for PET AC in terms of similarity and relative SUV differences to CTAC, chosen by the authors as the standard for AC. The usage of a deep learning approach normally involves long waiting times during training, which was circumvent by using a dedicated deep learning data centre GPU. However, for a full deployment as a scanner AC method, the training set would need to see its number at least doubled, also include data for patients with skull abnormalities, e.g. missing bone, 3D printed bone implants, etc… The proposed method is of high interest to all GE PET/MR users doing brain PET, as well as it opens a precedent for improvement in AC methods for all other PET/MR manufacturers.  Minor comments

 1. Line 33: Where is red Ladegofeld et al. should be Ladefoged et al.

Thanks for noticing this. Manuscript was updated accordingly.

 2. Line 77 – 80: authors explain registration of CT to ZTE but fail to state which interpolation method (trilinear, b-spline) was used. E.g. CT (512x512x71)-> ZTE (128x128x89).

4th degrees-B-spline interpolation was performed in SPM. Manuscript was updated accordingly. 

 3. Line 125: Jaccard Index is more sensitive to mismatch. Why did the authors decided to use Dice instead?

We choose the DSC index as it is the one that has been mostly reported in the previous study that allowed us to compare our CNN with the other techniques previously described.

 4. Having a table for Dice coefficients for the 3 methods, even as supplementary would be relevant for some readers.

Sadly Atlas AC-maps were not available. We nevertheless added in the supplementary a table with DSC and Jaccard coefficients for ZTE and Unet AC-maps. 

 5. Adding whole-brain grey matter to S1 Table would provide a better overall understanding of the results.

Whole brain gray matter cortical errors were added at the first line of S1 Table.

 6. Lines 68, 69,70,121: mm2 and mm3 should be superscript

Thanks for spotting this, manuscript was updated.

 7. Line 176 and 188: % character is missing in (3.7 ± 4.7) and (0.79, 3.13 and 0.86), respectively.

Thanks for spotting those mistakes. Manuscript was updated.

 8. How big is the impact of the calcification detection in a LOR passing though that area, how much is it relevant?

The calcification is anecdotic and impact is probably to be negligible. 

 9. Could the authors comment on the performance of the 3D-CNN if a subject shows skull abnormalities (e.g. missing skull)?

No patient in the training dataset nor in the validations datasets had history of brain surgery or skull trauma. It is really difficult in the context of deep learning to predict the transferability of its performance in unknown populations. Nevertheless, as shown with the small meningioma, we hope that the learned features will transfer well. 

In discussion was added P9/L261 : It is to note that as on no major morphological skull or brain abnormalities were present in the training nor in the validation datasets, it is difficult to infer how the CNN will behave in those conditions. Nevertheless, the small falci meningioma visible in Figure 1 is reassuring. Also, a previous study [ladegofed2019] including abnormal brain MRI with history of surgery and presence of titanium coils in 33% of patients which resulted to signal voids were well recovered by their deep learning algorithm.

---

## [Editor Report · Decision Letter 1]

16 Sep 2019

Attenuation Correction using 3D Deep Convolutional Neural Network for Brain 18F-FDG PET/MR: Comparison with Atlas, ZTE and CT based Attenuation Correction.

PONE-D-19-16876R1

Dear Dr. Blanc-Durand,

We are pleased to inform you that your manuscript has been judged scientifically suitable for publication and will be formally accepted for publication once it complies with all outstanding technical requirements.

With kind regards,

Stephen D. Ginsberg, Ph.D.

Section Editor

PLOS ONE

---

## [Editor Report · Acceptance letter]

27 Sep 2019

PONE-D-19-16876R1 

Attenuation Correction using 3D Deep Convolutional Neural Network for Brain ^18^F-FDG PET/MR: Comparison with Atlas, ZTE and CT based Attenuation Correction. 

Dear Dr. Blanc-Durand:

I am pleased to inform you that your manuscript has been deemed suitable for publication in PLOS ONE. Congratulations! Your manuscript is now with our production department. 

With kind regards,

on behalf of

Dr. Stephen D Ginsberg 

Section Editor

PLOS ONE